# Identifying Wild Versus Cultivated Gene-Alleles Conferring Seed Coat Color and Days to Flowering in Soybean

**DOI:** 10.3390/ijms22041559

**Published:** 2021-02-04

**Authors:** Cheng Liu, Xianlian Chen, Wubin Wang, Xinyang Hu, Wei Han, Qingyuan He, Hongyan Yang, Shihua Xiang, Junyi Gai

**Affiliations:** Soybean Research Institute & MOA National Center for Soybean Improvement & MOA Key Laboratory of Biology and Genetic Improvement of Soybean (General) & State Key Laboratory for Crop Genetics and Germplasm Enhancement & Jiangsu Collaborative Innovation Center for Modern Crop Production, Nanjing Agricultural University, Nanjing 210095, China; wildsoybean@163.com (C.L.); cxl005200@163.com (X.C.); 2015201023@njau.edu.cn (X.H.); hanwei2078125@163.com (W.H.); heqingyuan1@163.com (Q.H.); yhy8373@163.com (H.Y.); zgxiangshihua@163.com (S.X.)

**Keywords:** annual wild soybean (*G. soja* Sieb. and Zucc.), chromosome segment substitution line (CSSL), cultivated soybean (*G. max* (L.) Merr.), days to flowering, seed coat color, SNP linkage disequilibrium block (SNPLDB), whole-genome re-sequencing

## Abstract

Annual wild soybean (*G. soja*) is the ancestor of the cultivated soybean (*G. max*). To reveal the genetic changes from *soja* to *max*, an improved wild soybean chromosome segment substitution line (CSSL) population, *SojaCSSLP5*, composed of 177 CSSLs with 182 SSR markers (SSR-map), was developed based on *SojaCSSLP1* generated from *NN1138-2*(*max*)×*N24852*(*soja*). The *SojaCSSLP5* was genotyped further through whole-genome resequencing, resulting in a physical map with 1366 SNPLDBs (SNP linkage-disequilibrium blocks), which are composed of more markers/segments, shorter marker length and more recombination breakpoints than the SSR-map and caused 721 new wild substituted segments. Using the SNPLDB-map, two loci co-segregating with seed-coat color (SCC) and six loci for days to flowering (DTF) with 88.02% phenotypic contribution were identified. Integrated with parental RNA-seq and DNA-resequencing, two SCC and six DTF candidate genes, including three previously cloned (*G*, *E2* and *GmPRR3B*) and five newly detected ones, were predicted and verified at nucleotide mutant level, and then demonstrated with the consistency between gene-alleles and their phenotypes in *SojaCSSLP5*. In total, six of the eight genes were identified with the parental allele-pairs coincided to those in 303 germplasm accessions, then were further demonstrated by the consistency between gene-alleles and germplasm phenotypes. Accordingly, the CSSL population integrated with parental DNA and RNA sequencing data was demonstrated to be an efficient platform in identifying candidate wild vs. cultivated gene-alleles.

## 1. Introduction

Annual wild soybeans (*G. soja* Sieb. and Zucc.) are recognized as the wild progenitor of the cultivated soybeans (*G. max* (L.) Merr.) [1]. During the long term of artificial improvement, its classic characteristics of small seed, twining stem, pod shattering, and so on, were dramatically replaced with large seed, erect stem and pod non-shattering, but many desirable traits such as high protein content and adaptation to environmental stresses were weakened. Although the overall economic characters of wild soybean are inferior to those of the cultivated soybean, the potential of utilizing *G. soja* as a source of genetic diversity to improve the cultivated soybean, especially in coping with abiotic and biotic stresses, has been emphasized and demonstrated case by case [2,3]. Thus, it is expected to explore and transfer useful wild gene-alleles into soybeans through breeding programs.

In order to map the useful qualitative gene and quantitative trait locus (QTL) along with their corresponding alleles in wild soybeans, researchers have developed different kinds of genetic population, such as recombinant inbred line populations, doubled-haploid populations, natural populations, etc. However, due to the genetic background noise, these genetic populations are not efficient in identifying genes/QTLs individually, while chromosome segment substitution lines (CSSLs), as composed of multiple near-isogenic lines were potential in identifying individual genes/QTLs/segments after genetic noise removed. A CSSL population is generally developed through advanced backcrossing integrated with selfing and marker-assisted selection. To explore the favorable alleles deposited in the wild soybean, we had developed a CSSL population *SojaCSSLP1* by crossing a wild soybean *N24852* as donor with a cultivated soybean *NN1138-2* as receiver followed by continuously backcrossing to the latter integrated with SSR(Simple Sequence Repeat)-assisted selection. From *SojaCSSLP1*, improvement has been made, which led to the establishment of *SojaCSSLP2, SojaCSSLP3* and *SojaCSSLP4*. By using these CSSL populations, the wild alleles/haplotypes of genes/QTLs/segments associated with agronomic and seed quality traits were detected [4,5,6,7]. (To make the concept clear in the present study, three pairs of terms were used in gene and QTL mapping studies, i.e., segment (marker) and its haplotypes, qualitative gene and its alleles, QTL and its alleles; while candidate gene and its alleles were used in candidate gene analysis. Here “allele” was used in different cases because it was used in these ways already in literature. However, it can be distinguished from its context. In those cases of necessity, QTL-alleles, gene-alleles, candidate gene-alleles may be used.)

Genotyping-by-sequencing (GBS) is a high-throughput method that uses abundant SNP markers to ensure the accuracy of genotyping each CSSL and estimating the length of substituted segments in each CSSL [8]. However, due to the fact that the GBS often includes a relatively large proportion of missing segments and a small but hardly corrected percentage of SNP genotyping sequencing errors, the SNP linkage disequilibrium blocks (SNPLDBs), grouped from a segment of sequential SNPs separated at recombination sites across the whole population, is an option to solve the GBS limitations [9]. The previous research has demonstrated that the SNPLDB is more powerful in detecting QTL than that of traditional markers in natural populations [10], nested association mapping population [11] and recombinant inbred lines [12]. In addition, marker density is one of the key factors for mapping resolution. With high-density linkage map constructed from GBS, the fine mapping of QTL in rice [13], maize [14], soybean [15] and the other plants have been carried out.

Genetically, qualitative trait (such as seed coat color, SCC) and quantitative trait (such as days to flowering, DTF) may involve different kinds of inheritance mechanism. For SCC, the latest research showed that during the domestication process from the wild to the cultivated soybeans, the seed coat color changed from black to yellow might be due to an inversion at the *I* locus on chromosome (Chr.) 8 [16]. In cultivated soybean, the green seed coat was controlled by the gene *G* that has been cloned [17]. For DTF, it is an important ecological and agronomic trait. Up to now, a total of 13 genes including *E1*—*E10* [18,19,20,21,22,23,24,25,26], *J* [27], *Tof12* and *Tof11* [28] and 104 QTLs were identified for DTF or its linked trait DTM (days to maturity) (http://www.soybase.org/). Among them, seven genes, *E1* [29], *E2* [30], *E3* [31], *E4* [32], *J* [33] and *Tof12* and *Tof11* [28] have been cloned. *E1* has impact on DTF in soybean, and appears to be a legume-specific transcription factor that has a putative nuclear localization signal and a B3 distantly related domain [29]. *E2* is an ortholog of the Arabidopsis *GIGANTEA* (*GI*) gene to delay flowering and maturity [30]. *E3* and *E4* are two phytochrome A homoeologs, *GmPHYA3* and *GmPHYA2*, respectively [31,32]. *J* as the ortholog of Arabidopsis *EARLY FLOWERING 3* (*ELF3*) depends genetically on *E1* [33]. *Tof12 and Tof11* are two pseudo-response regulator genes and act via *LHY* homologs to promote expression of the *E1* gene and delay flowering under long photoperiods [28]. *E9* is *GmFT2a*, an ortholog of Arabidopsis *FLOWERING LOCUS T*. *GmFT4* is identified as the potential candidate gene for *E10* locus [26].

The present study was aimed at identifying the genetic differentiation of the two representative traits SCC and DTF between *G. soja* and *G. max*. At first, the study was designed to obtain an improved CSSL population *SojaCSSLP5* based on *SojaCSSLP1* established by using SSR markers, then (1) to densify the *SojaCSSLP5* with SNPLDB markers, (2) to identify the segments conferring the two traits, SCC and DTF based on the densified *SojaCSSLP5*, (3) to predict and verify the candidate genes conferring the two traits from the identified segments integrated with parental RNA-seq and DNA resequencing data and, finally, (4) to demonstrate the identified candidate genes using both *SojaCSSLP5* and germplasm accessions. Based on the obtained results, the evolutionary mechanism of the traits from the wild to the cultivated was discussed.

## 2. Results

### 2.1. Establishment of Densified Physical Map of SojaCSSLP5 with Genomic Marker SNPLDB

The improved CSSL population, *SojaCSSLP5,* composed of 177 lines with 182 SSR markers, in comparison to the previous *SojaCSSLP1-4* populations, the ratio of heterozygous segment was significantly reduced, accounting for only 0.02%, which may improve the accuracy of QTL mapping (Appendix A). 

As marker density is one of the key factors for precise mapping, the *SojaCSSLP5* and its two parents *NN1138-2* and *N24852* were sequenced with the average resequencing depths of 3.04, 11.39 and 9.51, respectively (Appendix A). By using the Williams82 v2.1 as the reference genome, a total of 2,567,426 SNPs were identified in *SojaCSSLP5*, accounting for 76.8% of the total SNPs (3,344,077) between the two parents (Appendix A). Most of the 2,567,426 SNPs were in the intergenic regions (88.3%), whereas the left SNPs (11.7%) were in the genic regions, including exons, splice sites, and untranslated regions (Appendix A), which indicated that high-density SNPs have been captured.

For a better comparability with the previous mapping results, the consecutive 2,567,426 SNPs were divided into 61,702 SNPLDBs based on 750 germplasm accessions (unpublished). According to the breakpoint information of each CSSL, a total of 1366 SNPLDB markers (or segments, including the flanking parts) were identified in *SojaCSSLP5*.

Based on the physical locations and genotypes of 182 SSRs and 1366 SNPLDBs, the lengths and locations of wild substitution segments in each CSSL of *SojaCSSLP5* were identified, from which two physical maps (SSR-map and SNPLDB-map) of the *SojaCSSLP5* were constructed as shown in Figure 1A and B. In the SNPLDB-map, the recovery percentage of *NN1138-2* genome in each CSSL was ranging from 77.50 to 99.50%, with an average of 95.33% in *SojaCSSLP5* (Figure 1C). The 177 CSSLs carried 1568 wild chromosome substitution segments, and each CSSL contained 1 to 24 wild substitution segments with an average of 8.8 segments (Figure 1D). Of these, five CSSLs carried only one wild segment, five carried two and 167 carried three or more wild substitution segments. The 1568 wild segments distributed on 20 chromosomes with a range of 41 on Chr. 18 to 199 on Chr. 6 (Table 1). The length of substitution segments in *SojaCSSLP5* ranging from 0.01 Mb to 52.50 Mb, with an average of 5.16 Mb for the homozygous wild segments (Figure 1E). The wild segments covered the entire wild soybean genome except four genomic regions of Gm03_LDB_25, Gm05_LDB_3, Gm07_LDB_14 and Gm07_LDB_18, hence the coverage of substitution segments were 99.79, 98.32 and 96.30% on Chr. 3, Chr. 5 and Chr. 7, respectively (Table 1).

In comparison to the SSR-map with 182 SSR markers, the SNPLDB-map with 1366 SNPLDB markers in *SojaCSSLP5* has the following advantages: (1) more wild segments were detected in *SojaCSSLP5*, including 721 newly identified wild segments and 84 broken-caused wild segments (Table 1, Figure 1). For example, in Figure 1F, a new wild segment was detected in *CSSL172* on Chr. 10 in SNPLDB-map, but no wild segment in SSR-map; based on SSR-map, one wild segment was detected in *CSSL080*, while the wild segment was broken into three segments in the SNPLDB-map, which might be due to double crossing happened. (2) The length of wild segments could be estimated more accurately, such as the average length of wild segments in SNPLDB-map (5.16 Mb) was about half of that (10.06 Mb) in SSR-map. (3) The SNPLDB-map could detect genomic regions misjudged as wild segments in SSR-map. For example, the coverages of wild segments were 100% in SSR-map while 99.74% in SNPLDB-map. That means 0.26% non-wild genome regions were misjudged as wild genome region in SSR-map (Table 1). The above results indicated that the genome composition of *SojaCSSLP5* could be accurately assessed by the densified SNPLDBs, which were obtained through whole genome resequencing. 

### 2.2. Identification of the SCC and DTF Wild Segments Using SNPLDB-Map 

#### 2.2.1. The Wild Segments Related to Seed Coat Color (SCC)

In *SojaCSSLP5*, the qualitative trait, SCC, was investigated and classified into three groups, including 23 black CSSLs (*B*), 7 green CSSLs (*G*) and 147 yellow CSSLs (*Y*), respectively, in a total of 177 CSSLs (Figure 2). The differences between CSSLs and *NN1138-2* in SCC must be due to the introgression of wild segments. Based on the comparisons of the genome composition between different SCC CSSL groups, the polymorphic SNPLDBs, Gm01_LDB_74 and Gm08_LDB_32, were detected between *Y* and *G* groups and between *Y* and *B* groups, respectively (Figure 2). The two segments/markers were found corresponding to the two reported SCC genes, *G* [17] and *I* [16,34] in the regions of 52,865,890–53,515,092 bp on Chr. 1 and 8,015,591–8,819,462 bp on Chr. 8, respectively (Table 2). The wild segments on Gm01_LDB_74 and Gm08_LDB_32, were associated completely with the phenotypes of green and black, respectively (Figure 2).

When using the SSR-map, we were failed to detect the marker that was co-segregated with the phenotype, although the wild segments of Sat_160 and AW132402 were associated with green seed coat and black seed coat, respectively (Figure 2). As shown in Figure 2, a total of 10 CSSLs contained the wild segment of Sat_160 in *SojaCSSLP5*, among which 3 and 3 CSSLs were with black and yellow seed coat, respectively. For AW132402, a total of 14 CSSLs contained the wild segment on the marker in *SojaCSSLP5*, among which 3 CSSLs were with yellow seed coat and the others with black seed coat. Meanwhile, 12 black seed coat CSSLs carried cultivated segment of AW132402. Therefore, the marker types and phenotypes may be associated but not necessarily co-segregated in SSR-map. In addition, the physical lengths in SNPLDB-map were much shorter than that in SSR-map (Table 2). The segment lengths of Sat_160 and AW132402 in SSR-map were about 2.28 Mb and 4.76 Mb, respectively, while the lengths of Gm01_LDB_74 and Gm08_LDB_32 in SNPLDB-map were only 0.65 Mb and 0.80 Mb, respectively (Table 2). Moreover, the markers Gm01_LDB_74 and Gm08_LDB_32 were identically co-segregated with the phenotypes of green seed coat and black seed coat in SNPLDB-map, but the markers Sat_160 and AW132402 were only accounted for 33 and 77% of CSSLs’ phenotypic variations in SSR-map, respectively (Table 2).

#### 2.2.2. The Wild Segments Associated with Days to Flowering (DTF)

In *SojaCSSLP5*, the quantitative trait, DTF, was evaluated under three environments. As shown in Appendix A, a broad phenotypic variation was observed between the cultivated and wild parents and among their *SojaCSSLP5* lines. The difference in DTF between *NN1138-2* and *N24852* was 13.0 d, while in *SojaCSSLP5*, the DTF varied in 50.0–64.0 d with an average of 53.7 d. The heritability of DTF was 95.6%, suggesting genetic variation accounting for a major part of the phenotypic variance in the population. The ANOVA showed that significant differences existed among the CSSLs (Appendix A).

The average values of DTF over environments were used for QTL mapping by using the ‘*csl*’ program in IciMapping software. Totally, three and six segments/markers/QTLs were identified on SSR-map and SNPLDB-map, respectively (Table 2, Figure 3). The 6 segments in SNPLDB-map, i.e., Gm04_LDB_41, Gm10_LDB_46, Gm12_LDB_16, Gm15_LDB_44, Gm16_LDB_1 and Gm17_LDB_62, explained 0.77, 67.05, 11.42, 2.26, 2.23 and 4.28% of phenotypic variation, respectively (Table 2). Among them, Gm10_LDB_46 had highest value of LOD (79.28) and percentage of phenotypic variation explained by individual QTL (PVE, 67.05%), and three loci, Gm10_LDB_46, Gm 12_LDB_16 and Gm15_LDB_44, had been reported by Watanabe et al. [30], Li et al. [35] and Pan et al. [12], respectively, while Gm04_LDB_41, Gm16_LDB_1 and Gm17_LDB_62 were newly detected QTLs in *SojaCSSLP5*. The wild alleles on Gm04_LDB_41, Gm15_LDB_44 and Gm16_LDB_1, showed negative additive effects ranging from −0.58 d to −1.00 d; the wild alleles on Gm10_LDB_46, Gm12_LDB_16 and Gm17_LDB_62 showed positive additive effects from 2.07 d to 3.68 d.

Comparing the QTL mapping results between SSR-map and SNPLDB-map, more QTLs (6 vs. 3) with higher PVE (88.02% vs. 49.25%) and shorter physical lengths (0.28 Mb of Gm10_LDB_46 vs. 2.30 Mb of satt243) were found in the latter (Table 2). In addition, there were 34 CSSLs significantly different from the recurrent parent *NN1138-2* in DTF, including 24 later flowering and 10 earlier flowering CSSLs (Figure 3). According to the mapping results of SNPLDB-map, the wild segments of Gm10_LDB_46, Gm12_LDB_16 and Gm17_LDB_62 could explain the phenotypic variations of all the later flowering CSSLs, including 16, 6 and 4 CSSLs, respectively; the wild segments of Gm04_LDB_41, Gm15_LDB_44 and Gm16_LDB_1 could explain the phenotypic variations of all the early flowering CSSLs except CSSL123. However, the segments detected in SSR-map, Satt243, Satt488 and Sat_163 located on Chr. 10, Chr. 17 and Chr. 18, could only explain 12, 2 and 4 (17 in 34) CSSLs’ phenotypic variation, respectively. In a word, the wild segments of DTF detected by SNPLDB-map could explain all the 34 CSSLs’ phenotypic variation except CSSL123, but the wild segments detected by SSR-map could explain only 17 CSSLs’ phenotypic variation in DTF. Furthermore, SNPLDB-map could not only detect the positive wild alleles but also negative wild alleles; while the wild alleles of DTF detected in SSR-map were all positive. The above results indicated that by using the densified SNPLDB-map, the DTF QTLs could be fully explored in a smaller genomic interval at a higher accuracy in comparison to using the SSR-map, therefore, the former is more potential in identifying the QTLs/segments.

Taken together, the above results indicated that the mapping accuracy and resolution of the densified SNPLDB-map is much higher than that of the primary SSR-map; thus, the *SojaCSSLP5* with new developed SNPLDB-map can be used to analyze the genetic base of qualitative and quantitative traits, such as SCC and DTF.

### 2.3. Prediction and Primary Verification of SCC and DTF Candidate Genes from the Identified Segments

The basic idea in prediction and primary verification of candidate genes of SCC and DTF were (1) to find all the genes in the identified segment according to SoyBase (http://www.soyase.org), (2) to choose the possible candidate genes according to the two parent’s RNA-seq data in a set of different tissues, (3) to find the most possible candidate genes according to the sequence differences by using high depth genome resequencing of the two parents, and (4) to annotate the predicted candidate genes based on Gene Ontology (GO) analysis.

#### 2.3.1. The Candidate Genes Related to Seed Coat Color (SCC)

In searching the candidate gene of *G*, the linked segment of Gm01_LDB_74 was between 52,865,890 bp to 53,515,092 bp on Chr. 1, in which 81 genes contained according to SoyBase (http://www.soybase.org). Among them, *Glyma.01G198500* performed significantly different expressions with more than two-fold difference between *N24852* and *NN1138-2* among four tissues, including flower, 14 seed, 35 seed and 21 Pod (Table 3, Appendix A). Meanwhile, the genome sequences of *Glyma.01G198500* was different between the two parents with the nucleotide G in black seed coat *N24852* mutated to A in the yellow seed coat *NN1138-2* in intron region, which led to an alternative splicing site and generated a premature stop codon (Table 4, Figure 4A, Appendix A). The above results were consistent with Wang et al. [17], who have cloned and verified *G* gene (*Glyma.01G198500*), encoding the *CAAX* amino terminal protease protein.

As for the *I* locus, the linked segment of Gm08_LDB_32 was between 8,015,591-8,819,462 bp on Chr. 8, in which 108 genes contained according to SoyBase (http://www.soybase.org). Among them, nine key chalcone synthase genes may be responsible for SCC. The gene expression data showed that three of them, *Glyma.08G109200*, *Glyma.08G109300* and *Glyma.08G109400* performed high expression levels in different stages of seed development (flower, 14seed, 21seed, 28seed and 35seed). However, only *Glyma.08G109400* showed the highest expression values in all the five tissues, especially performed significantly differential expressions between *N24852* and *NN1138-2*, even more than two-fold in the eight tissues except in 7pod (Table 3, Appendix A). In addition, only *Glyma.08G109400* had a nonsynonymous mutation on 8,392,915 bp, in which the nucleotide G of the black seed coat *N24852* mutated to A of the yellow seed coat *NN1138-2* (Table 4, Figure 4B, Appendix A). Hence, the mutation on 8,392,915 bp of *Glyma.08G109400* might be the main cause of the change in seed coat color between the two parents.

#### 2.3.2. The Candidate Genes Related to Days to Flowering (DTF)

For DTF, the three early flowering wild segments, on Gm04_LDB_41, Gm15_LDB_44 and Gm16_LDB_1, contained 31, 321 and 61 (419 in total) candidate genes, respectively, according to SoyBase (http://www.soybase.org). After the low expression genes with FPKM (Fragments Per Kilobase per Million) value <2.0 removed, the remained 156 candidate genes were annotated and the sequence differences between the two parents were compared (Appendix A). For Gm04_LDB_41, the candidate gene *Glyma.04G167900*, which was responded to light stimulus, performed significantly different expressions with more than two-fold difference between *N24852* and *NN1138-2* in four tissues, including leaf, 14seed, 28seed and 35seed (Table 3, Appendix A). The sequence analysis showed that 11 polymorphic SNPs were detected in promoter region of *Glyma.04G167900* between two parents, with no SNP difference on exons (Table 4, Figure 4C). For Gm15_LDB_44, *Glyma.15G221300* may be a most likely candidate gene, because it not only had a nonsynonymous mutation on 39,953,287 bp with T in *N24852* mutated to G in *NN1138-2*, but also it showed significantly different expressions of more than two-fold between *N24852* and *NN1138-2* in six tissues, including leaf, 14seed, 21seed, 28seed, 35seed and 21pod (Table 3 and Table 4, Figure 4D, Appendix A). A candidate gene of Gm16_LDB_1, *Glyma.16G005100* showed significantly different expressions of more than two-fold between *N24852* and *NN1138-2* in five tissues, including leaf, flower, 14seed, 7pod and 21pod (Table 3 and Appendix A). Meanwhile, one missense and one stop lost variants happened on 349,809 bp and 349,961 bp, respectively, resulted in T and G in *N24852* mutated to G and T in *NN1138-2* (Table 4, Figure 4E).

For the later flowering wild segments, Gm10_LDB_46 was mapped into the 277.5 kb genomic interval from 45,288,662 bp to 45,566,206 bp on Chr.10, in which the *E2* gene was reported by Watanabe et al. [30]. The function gene of *E2* was *Glyma.10G221500*, which was an important photoperiod regulating gene. According to the parents’ sequences of *Glyma.10G221500*, there were eight SNPs in introns, but they did not change coding sequence. While the nucleotide A on 45,310,798 bp from the late flowering wild soybean *N24852* was mutated to T in early flowering *NN1138-2*, which resulted in a premature stop codon (Table 4, Figure 4F, Appendix A). Meanwhile, *Glyma.10G221500* performed significantly different expressions of more than two-fold difference between *N24852* and *NN1138-2* in four tissues of leaf, flower, 28seed, and 21pod (Table 3 and Appendix A). These results were consistent with those reported by Watanabe et al. [30]. 

Gm12_LDB_16 was mapped into the 48.7 kb genomic interval from 5,497,551 bp to 5,546,301 bp on Chr.12, which contained five candidate genes according to SoyBase (http://www.soybase.org). Through gene annotation and expression analysis, *Glyma.12G073900* may be a candidate gene, because it showed significantly different expressions between *N24852* and *NN1138-2* in five tissues, i.e., leaf, 14seed, 21seed, 7pod and 21pod (Table 3 and Appendix A). The protein sequence of this gene was similar to the *PRR3* gene in *Arabidopsis*. The encoding protein of *PRR3* could bind to CO (CONSTANS) protein to delay crop flowering. Li et al. [35] reported some functions of this gene. As *Glyma.12G073900* may be the most likely candidate gene of DTF, we also cloned this gene from the two parents, which indicated that the C and C nucleotide in the wild soybean *N24852* was mutated to A and T in the cultivated soybean *NN1138-2*, respectively, which generated a nonsynonymous mutation and a premature stop codon (Table 4, Figure 4G, Appendix A).

The new DTF QTL/SNPLDB, Gm17_LDB_62, was mapped into the 132 kb genomic segment from 40,650,273 to 40,782,338 bp on Chr. 17, in which 19 candidate genes contained. Among them, *Glyma.17G253700* showed significantly different expressions between *N24852* and *NN1138-2* in three tissues of leaf, 21seed, and 35seed (Table 3 and Appendix A). Meanwhile, the sequences of *Glyma.17G253700* were different between *N24852* and *NN1138-2* with the nucleotides A and T in *N24852* mutated to T and C in *NN1138-2* on 40,762,395 bp and 40,766,099 bp, respectively, which both resulted in missense variants (Table 4, Figure 4H, Appendix A). The annotations indicated that *Glyma.17G253700* encoded UDP-Glycosyltransferase superfamily protein which is related to flowering in *Arabidopsis thaliana* [36]. Thus, *Glyma.17G253700* might be the candidate gene of Gm17_LDB_62.

In summary, combined the segment identification using a SNPLDB-map with parental RNA-seq and high depth whole genome resequencing data, eight candidate genes (*Glyma.01G198500*, *Glyma.08G109400*, *Glyma.04G167900*, *Glyma.10G221500*, *Glyma.12G1073900*, *Glyma.15G221300*, *Glyma.16G005100* and *Glyma.17G253700*) were predicted from two SCC segments (Gm01_LDB_74 and Gm08_LDB_32) and six DTF segments (Gm04_LDB_41, Gm10_LDB_46, Gm12_LDB_16, Gm15_LDB_44, Gm16_LDB_1 and Gm17_LDB_62), respectively (Table 4). 

### 2.4. Demonstration of SCC and DTF Candidate Genes from Allele-Phenotype Coincidence in SojaCSSLP5

To verify that these gene mutations were correct and universal, the allelic variations of these candidate genes in *SojaCSSLP5* were scanned, in which 7 and 23 lines carrying wild haplotypes D1 and E1 in *Glyma.01G198500* and *Glyma.08G109400* were all with green and black seed coat, respectively (Table 5), while 147 lines with cultivated haplotype D2E2 were yellow seed coat. This indicated that the consistencies between haplotype/allele and phenotype were both 100% and SCC in *SojaCSSLP5* was controlled by two genes, *Glyma.01G198500* and *Glyma.08G109400*. 

For DTF, as shown in Table 5, F1 and F2, H1 and H2, I1 and I2, J1 and J2, K1 and K2, L1 and L2, represent the wild *N24852* and cultivated *NN1138-2* haplotypes of *Glyma.04G167900*, *Glyma.10G221500*, *Glyma.12G073900*, *Glyma.15G221300*, *Glyma.16G005100* and *Glyma.17G253700*, respectively. In comparison with the 143 CSSLs with the cultivated genotype of F2H2I2J2K2L2, the wild haplotypes F1 in *Glyma.04G167900*, J1 in *Glyma.15G221300* and K1 in *Glyma.16G005100* on the genetic background of H2I2J2K2L2, F2H2I2K2L2 and F2H2I2J2L2 could significantly shorten DTF 1.8, 2.2 and 1.8 days, respectively (Table 5). Meanwhile, 13, 5, and 2 CSSLs carrying the wild haplotypes H1 in *Glyma.10G221500*, I1 in *Glyma.12G1073900*, and L1 in*Glyma.17G253700* on the genetic background of F2I2J2K2L2, F2H2J2K2L2 and F2H2I2J2K2, respectively, showed 7.8, 6.3 and 5.0 days longer in DTF (Table 5). These results indicated that in *SojaCSSLP5*, the wild and cultivated haplotypes of the eight genes indeed have different effects on DTF.

### 2.5. Further Demonstration of the Candidate Gene-Allele Effects in Germplasm Accessions 

In germplasm accessions, six of the eight detected candidate genes, *Glyma.01G198500*, *Glyma.10G221500*, *Glyma.12G073900*, *Glyma.15G221300*, *Glyma.16G005100* and *Glyma.17G253700* were detected with their haplotypes consistent to *N24852* and *NN1138-2*, which were presented by D1 vs. D2, H1 vs. H2, I1 vs. I2, J1 vs. J2, K1 vs. K2, and L1 vs. L2, respectively (Table 5). A total of 303 cultivated accessions (44 green and 259 yellow SCC accessions) carrying the SNPs consistent to *N24852* and *NN1138-2* in all six genes were screened out to demonstrate the realness of the six genes.

For SCC, as shown in Table 5, the *N24852* haplotype D1 in *Glyma.01G198500* was associated with green SCC in all 44 green SCC soybeans, and the *NN1138-2* haplotype D2 in *Glyma.01G198500* was associated with yellow SCC in all the 259 yellow SCC cultivated soybeans, implying that *Glyma.01G198500* was the functional gene responding to SCC in the germplasm population.

For DTF, since the multiple genes of a same trait DTF may result in diverse genetic background which cause phenotypic interference, the accessions with a same genetic background as *NN1138-2* except for the accessing loci were picked up from the 303 accessions for further demonstrating the effects of *N24852* and *NN1138-2* haplotypes (Table 5). For *Glyma.10G221500*, *Glyma.12G073900* and *Glyma.17G253700*, the wild haplotypes H1, I1 and L1 on I2J2K2L2, H2J2K2L2 and H2I2J2K2 genetic background carried by 11, 7 and 14 accessions showed significantly delayed DTF 10.8, 8.2 and 10.6 days, respectively, comparing to the 38 accessions with *NN1138-2* haplotype H2I2J2K2L2 **(**Table 5). For *Glyma.15G221300* and *Glyma.16G005100*, 6 and 47 accessions carrying *N24852* haplotypes J1 and K1 on H2I2K2L2 and H2I2J2L2 genetic background caused significantly DTF 4.6 and 3.6 days shorter comparing to the 38 accessions with *NN1138-2* haplotype H2I2J2K2L2, respectively (Table 5). 

These results further demonstrated that the gene-alleles/haplotypes of SCC and DTF detected in *SojaCSSLP5*, as well as in their parents *N24852* and *NN1138-2*, have the same functions in the Chinese soybean germplasm population, therefore, these gene-alleles/haplotypes are correct and universal.

## 3. Discussion

### 3.1. Comparisons of the Present SCC and DTF Mapping Results with Those in the Literature

For SCC, at least six loci (G, I, T, W1, R, and O) and nine putative genes were reported [37]. However, only two loci, Gm01_LDB_74 and Gm08_LDB_32, corresponding to *G* and *I*, respectively, were detected to have allelic differentiation between the wild and cultivated parents in the present study. For the *G* locus related to green SCC, *Glyma.01G198500* in Gm01_LDB_74 was predicted to be the candidate gene, which had been cloned by Wang et al. [17]. For the *I* locus related to black SCC, in comparison to the previous results, Gm08_LDB_32 was mapped into a much smaller region than that of *SCC2-1* [34], *SCC3-1* [38] and *SCC4-1* [39], in which *Glyma.08G109400* was found as its candidate gene. As the previous results indicated, there should be five seed coat colors in hybrid progenies of *max*×*soja*, but in *SojaCSSLP5* only yellow, green and black SCC, no brown and bi-color SCC turned out. This might be due to the fact that the wild alleles related to the latter two colors lost randomly during the continuously backcrossing process in developing the *SojaCSSLP5*. Therefore the results on the gene system of SCC in *max* × *soja* has not been completed yet and should be studied further.

Regarding DTF, totally 104 putative QTLs were identified and recorded in SoyBase (http://www.soybase.org). Among these, seven genes, i.e., *E1* [29], *E2* [30], *E3* [31], *E4* [32], *J* [33] and *Tof12* and *Tof11* [28] have been cloned, but not by wild soybean. In the present study, a total of 6 DTF wild/cultivated segments/QTLs were identified. In comparison to the previous results, the segments of Gm10_LDB_46 and Gm15_LDB_44 were detected in both *max* × *soja* and *max* × *max* crosses [12,30,40], which might be the ones with allelic differentiation not only between *max* and *soja* but also among *max* and might be the key loci to explain the domestication and improvement of soybean. The segment of Gm12_LDB_16 was also detected in *max* × *soja* cross before [40], while the other three DTF QTL/segments were newly identified in *SojaCSSLP5*. Even though, in the CSSL, population may lost randomly some segments due to the continuous backcrossing, there are still three more wild segments newly identified for DTF. That means the genetic structure of *soja* is quite different from *max*, especially for DTF, and the wild soybean CSSL population is potential in exploring the genetic difference between the two species.

### 3.2. The Superiorities of SojaCSSLP5 with SNPLDB-Map and Its Potentials in Studying G. Soja Genome

In the present study, two maps, SNPLDB-map and SSR-map, were established for the same CSSL population *SojaCSSLP5* with the major difference lies in densified genomic marker SNPLDB in the former while limited number of PCR-based SSR marker in the latter. The results showed that the former was much superior over the latter in the following aspects: (i) More genome-wide SNPLDB markers were obtained, resulting in shorter segment length (5.16 Mb in average) and more detected wild segments (1568 in SNPLDB-map vs. 763 in SSR-map), especially making the double crossing segments (84) detectable. (ii) The genomic marker made a direct annotation of candidate genes for the detected segments, which led to a direct detection of the mutant nucleotide(s) when integrated with parental RNA-seq and DNA resequencing data.

The present results indicated that the wild CSSL population is efficient and effective in identifying segment/QTL/gene with genetic background noise removed by continuous backcrossing and the marker density densified by whole genome resequencing. Our final target CSSL population was a single wild chromosome segment substitution line population in which each line should be a near-isogenic line to the recurrent parent. However, the present *SojaCSSLP5* is in fact a multiple wild chromosome segment substitution line population, in which each line contains multiple wild segments. From the present study, we found that this kind of multiple wild chromosome segment substitution line population is also useful and potential. In which, multiple or group of lines could be compared and analyzed rather than only single line to be used to identify the target wild segment. The other advantage of this kind CSSL population also lies in that the involved lines can be backcrossed to the recurrent parent to obtain a secondary segregating population on the target segment from which fine mapping of the target gene can be done. In addition, in CSSL population, there might be some problem in random loses of some small segments during the continuously backcrossing process, especially due to the inadequacy of marker numbers in developing the CSSLs, and the multiple segments CSSL may be a compensation to it. 

### 3.3. The CSSL Population Integrated with Parental RNA-Seq and DNA Resequencing and Germplasm Scanning as a Platform in Studying Evolutionary Mechanism

The evolution from wild soybeans to cultivars has gone through a long history, but little is known about the evolutionary mechanism of a gene in domestication process. We used the wild soybean CSSL population to identify two and six segments, then integrated with 5× parental RNA-seq of eight different tissues and 10× parental genome resequencing, 2 SCC and 6 DTF candidate genes were identified and primary-verified. These genes were associated with one qualitative trait SCC, and one quantitative trait DTF, respectively, from which their allelic variations of wild vs. cultivated soybean explored. Then, the candidate genes were demonstrated with the allele-phenotype coincidence in *SojaCSSLP5* and further demonstrated with the 303 germplasm accessions. It indicated that the quantitative trait had more candidate genes and allelic variations than the qualitative trait. A single site nucleotide mutation may cause the allele change from the wild to the cultivated, especially for a qualitative trait; while multiple sites of nucleotide mutations also happened, especially for quantitative trait (Figure 4). Thus, nucleotide mutation, especially nonsynonymous mutation, could cause the evolution from the wild to the cultivated soybean at a single gene level. The *SojaCSSLP5* has provided the material bases for gene evolution research at a single gene level. However, after we looked at the genetic structure of a single gene in the germplasm population, we found that multiple allelic variations may happen on a locus, some cultivated nucleotides in *SojaCSSLP5* are a real evolved new nucleotides of cultivated soybean, but some may exist in wild soybean already. Thus, the alleles thought to be evolved in cultivated soybean should be checked in the germplasm population for exploring the real evolutionary mechanism of a gene. In any event, to find the linked segment in a *SojaCSSLP5* is the first step, segment identification integrated with parental RNA-seq and DNA resequencing along with germplasm demonstration composed a potential platform in identifying an evolutionary gene and evolutionary mechanism from *G. soja* to *G. max*.

## 4. Materials and Methods

### 4.1. Plant Materials and Phenotype Evaluation

An improved CSSL population *SojaCSSLP5* was established based on *SojaCSSLP1* through SSR-assisted selection. Its original parents were *NN1138-2* (*G. max*) and *N24852* (*G. soja*); the former was an elite cultivar in maturity group V, characterized with early flowering, yellow seed coat, used as recurrent parent; the latter was an annual wild soybean in maturity group III, characterized with late flowering, black seed coat, used as donor parent. We initially used 850 SSR markers from a high density integrated soybean genetic linkage map to survey the polymorphism between *NN1138-2* and *N24852*, and 151 polymorphic markers were selected for marker-assisted selection of target substituted segments [4]. Through continuous backcross followed with alternation of backcross, selfing and two generations of marker assisted selection, the first CSSL population *SojaCSSLP1* composed of 151 lines was developed by Wang et al. [4] in 2013. In the present study, to reduce the heterozygous segments and off-type lines, the *SojaCSSLP1* was selfed and SSR-assist-selected again to establish *SojaCSSLP5* with more homozygous lines. Finally, *SojaCSSLP5* was composed of 177 CSSLs with the wild genome separated into 177 diagonal segments by 182 SSR markers. Among the 177 CSSLs, there were 23, 12, 3, 62, 3, 25, 14, 21 and 14 CSSLs at the generations of BC_3_F_8_, BC_3_F_7_, BC_3_F_6_, (BC_2_F_3_)BC_1_F_6_, (BC_2_F_2_)BC_2_F_6_, BC_4_F_7_, BC_5_F_6_, BC_4_F_7_ and BC_5_F_6_, respectively. Among the 177 CSSLs, there were 138 lines derived directly from the *SojaCSSLP1*.

The *SojaCSSLP5,* along with its two parents, was planted in a complete randomized block design experiment with 3 replications, one row per plot, 1 m length, 10 plants per row and 0.5 m row space at Jiangpu Experiment Station, Nanjing, China (N31°02′, E118°04′) in 2016, and Dangtu Experiment Station, Maanshan, China (31°32′ N, 118°37′ E) in 2017 and 2018. The three environments were coded as 2016JP, 2017DT and 2018DT, respectively, and were all belonged to short-day growth conditions. A total of two traits, SCC and DTF were investigated on a plot basis. DTF was scored as the days from sowing to first flowering, which corresponded to the R1 developmental stage [41].

### 4.2. DNA Extraction and SSR-Map Construction

DNA was extracted from fresh leaves of each CSSL using the CTAB method [42] with minor modifications. A total of 182 SSRs that showed polymorphism between *NN1138*-*2* and *N24852* were used to genotype the *SojaCSSLP5*. The polymerase chain reaction (PCR) was conducted according to Panaud et al. [43] with minor modifications. The gels were run in PAGE running buffer (1 × TBE) at 200 V for 1.2 h and then silver-stained.

According to the position of each SSR marker, the SSR-map of *SojaCSSLP5* was constructed, covering 872.40 Mb of the soybean genome with an average distance of 4.64 Mb between neighboring markers according to the integrated physical map by Song et al. [44] and the reference genome of Willimas82.a2.v1 [45].

### 4.3. SNP Identification and SNPLDB-Map Construction

To accurately assess the genome composition for improving the QTL mapping resolution, whole genome re-sequencing was carried out for *SojaCSSLP5*. A paired-end sequencing library with an insert size of 350 bp was constructed for each CSSL and parent. First, 350 bp DNA fragments were randomly generated for each material using a hydrodynamic shearing platform (Covaris, https://covaris.com/ and ApplicationSupport@covaris.com). Next, the DNA fragments were treated according to the following manufacturer’s specifications (Illumina): fragments were end repaired, polyA-tails were added and they were ligated to paired-end adaptors and amplified by PCR. Re-sequencing was performed using the Illumina Hiseq^TM^PE150 platform in Novogene Bioinformatics Institute (Beijing, China).

The clean paired-end reads were mapped to the reference genome of Willimas82.a2.v1 [45] with Burrows-Wheeler Aligner (BWA) [46] and Sequence Alignment/Map tools (SAMtools) [47] to call SNPs. Then the SNPs in progenies were filtered according to the following threshold: minQ ≥ 30, max-missing ≤ 0.2 and min-mean DP ≥ 1.5 by VCFtools. 

To make the mapping results comparable among different populations, the consecutive SNPs in *SojaCSSLP5* were grouped into SNP linkage disequilibrium blocks (SNPLDB) based on 750 germplasm accessions (unpublished) using the software PLINK [48]. According to the breakpoint information of each CSSL, the co-segregation blocks were combined into a single unit as one SNPLDB. Accordingly, the SNPLDBs were used as genomic markers to construct a physical map, SNPLDB-map, for *SojaCSSLP5* using the GGT32 software [49]. In this case, the SSR-map and SNP-map were established for a same CSSL population *SojaCSSLP5*. 

### 4.4. Identification of Segment for SCC and DTF

For SCC, the segments were identified using bulk segregant analysis. The 177 CSSLs were grouped according to their SCC phenotypes (green, black and yellow), and the allele frequency differences between the bulks were tested. The markers/segments polymorphic between bulks might be related to SCC. For DTF, the segment/QTL analysis was performed using IciMapping V4.1 software [50] with the model of RSTEP-LRT-ADD (Stepwise regression based likelihood ratio tests of additive QTL). The LOD threshold was set to 2.5. To examine the association between phenotype and genotype, the segments/QTLs were verified by using the consistency value between the phenotype and genotype, *C*_P&G_ = *n_pg_*/*n_g_* × 100%, where *n_pg_* is the number of lines that their phenotypes are co-segregating with genotypes and *n_g_* is number of lines containing the detected wild segment.

### 4.5. Candidate Gene Annotation and Verification through RNA-Seq for SCC and DTF

According to SoyBase (http://www.soybase.org), the possible genes in the identified segments were annotated. To verify the annotated genes related to SCC and DTF, transcription analysis was conducted. A total of eight tissues, including young leaf, blooming flower, 14seed (seed at 14 days after flowering), 21seed, 28seed, 35seed, 7pod (pod at 7 days after flowering), 21pod, were harvested for *NN1138-2* and *N24852*, respectively. The whole genome gene expressions of these tissues were investigated through RNA-seq by Shanghai Biotechnology Corporation (Shanghai, China). Total RNA was extracted using TRIzol reagent by following the manufacturer’s protocols (Invitrogen, Carlsbad, CA, USA). RNA quality was inspected by an Agilent 2100 Bioanalyzer (Agilent technologies, Santa Clara, CA, US) to meet the criteria of OD_260/280_ ≥ 1.8, 28S/18S ≥ 1, and RNA Integrity Number (RIN) ≥ 7. One μg of RNA was used for paired-end library construction using Illumina TruSeq RNA Sample Preparation Kit (Illumina, San Diego, CA, USA). The RNA-seq was generated by the Illumina HiSeq 2000. To obtain clean reads, the raw reads were filtered via Seqtk (http://github.com/lh3/seqtk). The RNA-Seq reads from each sample were aligned to the Glycine max genome of the Williams82 v2.1 via the TopHat (version 2.0.9) [51] with the default options. The transcripts of each sample were reconstructed using String Tie (version 1.3.0) [52]. Then, the gene expression level was measured by the value of fragments per kilobase million (FPKM). Meanwhile, the sequence differences of candidate genes between the two parents were compared according to the resequencing data for NN1138-2 and N24852, respectively. To clone and compare the CDS of *Glyma.12G073900*, the gene-specific primers were designed based on the cDNA sequence of Williams82 v2.1, and used for PCR amplification of each homologous gene from *NN1138-2* and *N24852*. The primers were: 5′TCAAGTGCTTGGGATGTGGA3′, 5′GCCGTACTTCATCTGTCCCG3′.

### 4.6. Demonstration of SCC and DTF Candidate Genes through Allele-Phenotype Coincidence in SojaCSSLP5

To demonstrate the predicted SCC and DTF candidate genes, the resequencing data of SojaCSSLP5 were used to identify the haplotypes of the 8 candidate genes. Based on the nucleotide changes of the candidate genes between N24852 and NN1138-2, the haplotypes/alleles were scanned in SojaCSSLP5. D1 and D2, E1 and E2, F1 and F2, H1 and H2, I1 and I2, J1 and J2, K1 and K2, L1 and L2 represent the haplotypes of the eight verified candidate genes, including Glyma.01G198500, Glyma.08G109400, Glyma.04G167900, Glyma.10G221500, Glyma.12G073900, Glyma.15G221300, Glyma.16G005100 and Glyma.17G253700, for wild soybean N24852 and cultivated soybean NN1138-2, respectively. The gene-alleles were demonstrated through testing the coincidence between alleles and their phenotypic values.

### 4.7. Further Demonstration through Allele-Phenotype Coincidence in Chinese Soybean Germplasm Population

The Chinese soybean germplasm population is composed of 750 wild and cultivated accessions with its resequencing 5× depth data available. Out of the 8 candidate genes, only 6 genes (*Glyma.01G198500*, *Glyma.10G221500*, *Glyma.12G073900*, *Glyma.15G221300*, *Glyma.16G005100* and *Glyma.17G253700*) were detected the same allele/haplotype pairs of the two parents *N24852* and *NN1138-2*. In fact, not only two (*N24852* and *NN1138-2*) alleles but multiple alleles existed on the 6 loci in the germplasm population. In order to eliminate the interference caused by other alleles (or SNPs at other sites), the accessions carrying the alleles/haplotypes consistent with *N24852* and *NN1138-2* on the 6 candidate genes were selected for the demonstration analysis. The accessions with black seed coat were excluded because the corresponding candidate gene *Glyma.08G109400* was not included. Accordingly, 303 cultivated accessions were used to compare their phenotypes for further demonstration of the coincidence between alleles and their phenotypes, the compared two sets of materials having the respective parental alleles on each locus, with the other loci for a same trait all consistent, especially for DTF. The phenotypic data of SCC and DTF of the compared germplasm accessions were observed in a complete randomized block design experiment with 3 replications at Dangtu Experiment Station in 2018. The Student’s *t*-test was used to detect their significant differences.

## 5. Conclusions

Wild soybean (*G. soja*), characterized with small and black seed, twining stem, pod shattering, etc. is acknowledged to be the wild progenitor of the cultivated soybean (*G. max*). To reveal the genetic changes from *G. soja* to *G. max*, an improved wild soybean CSSL population *SojaCSSLP5* with segment heterozygosity reduced and both SSR and SNPLDB markers genotyped was established based on the previous versions obtained through advanced backcrossing. Compared to the low density of SSR-map, the highly densified SNPLDB-map could identify more wild segments with shorter length in *SojaCSSLP5* (1366 vs. 182 markers and 5.06 vs. 10.06 Mb/marker). By using *SojaCSSLP5* SNPLDB-map, two markers co-segregating with SCC and six markers associated with DTF with 88.02% PVE were mapped, among which three DTF QTLs were newly detected in the wild soybean. Integrated with parental RNA-seq and DNA resequencing data, two and six candidate genes were predicted based on allele-phenotype coincidence in *SojaCSSLP5* for SCC and DTF, respectively. Among these, one SCC candidate gene and four DTF candidate genes were newly reported here that might be used to broaden the genetic base of cultivated soybeans. The present results indicated that the mapping resolution of QTLs and qualitative genes was highly improved by using high density SNPLDB-map, and this kind of CSSL-map, if integrated with multiple allele information in germplasm population, is an potential platform in identifying candidate wild vs. cultivated gene-alleles and exploring evolutionary mechanism from wild to cultivated soybeans.

## Figures and Tables

**Figure 1 ijms-22-01559-f001:**
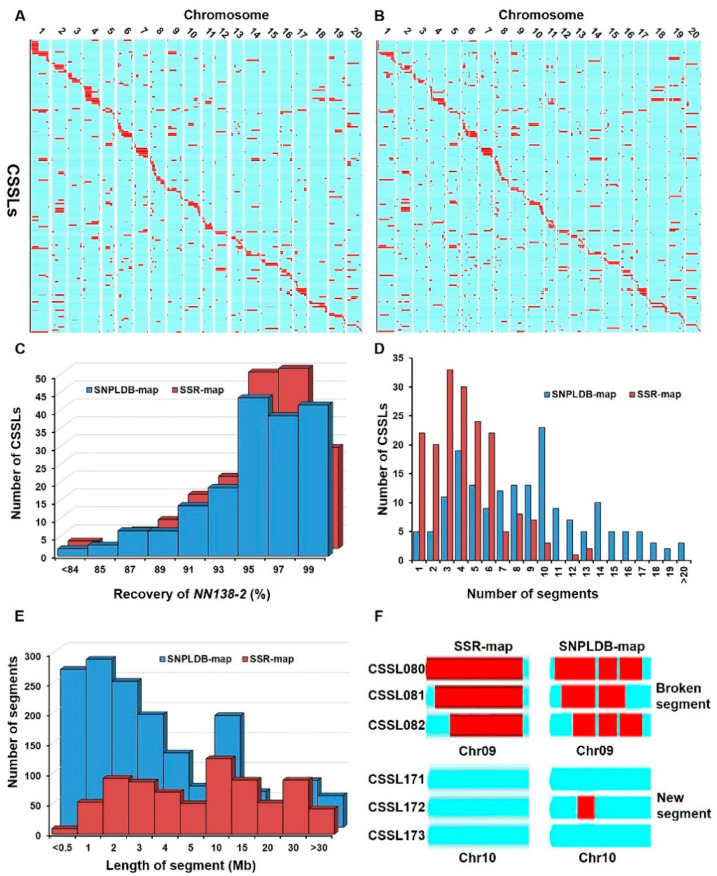
Molecular characteristics of *SojaCSSLP5* derived from *NN1138-2* and *N24852*. (**A**) The physical map of *SojaCSSLP5* genotyped by 182 SSR markers (SSR-map). (**B**) The physical map of *SojaCSSLP5* genotyped by 1366 SNPLDB markers (SNPLDB-map). (**C**) The frequency distribution of *NN1138-2*-recovery rate in *SojaCSSLP5.* (**D**) The frequency distribution of wild segment number per line in *SojaCSSLP5.* (**E**) The frequency distribution of wild segment length in *SojaCSSLP5.* (**F**) The SSR-segment broken into new SNPLDB segments. In (**A**,**B**,**F**), the light turquoise and red bars denote the *NN1138-2* and *N24852* segments, respectively.

**Figure 2 ijms-22-01559-f002:**
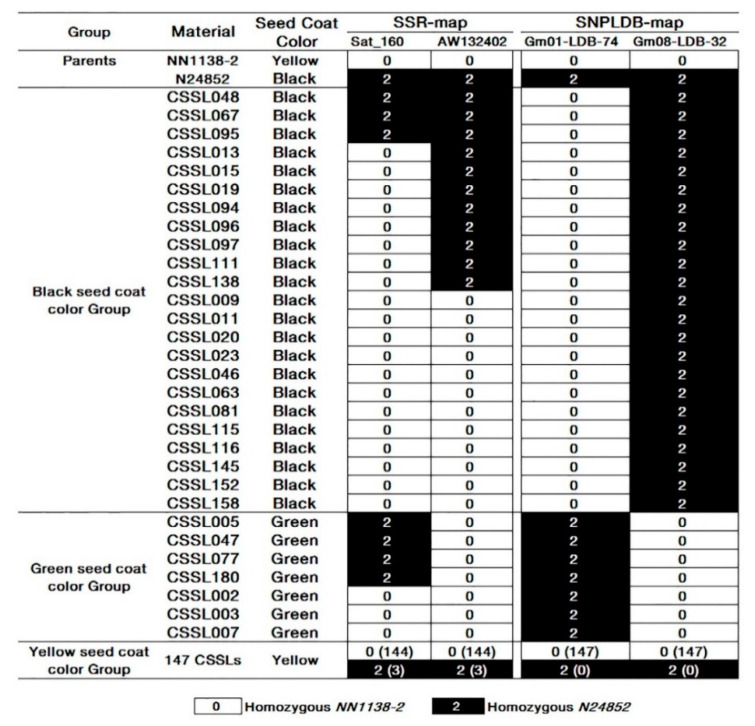
Identification of wild alleles for seed coat color using bulk segregant analysis. SSR-map, the physical map constructed with SSR markers; SNPLDB-map, the physical map constructed with SNPLDB markers. The seed coat color (SCC) was classified as yellow, green and black in *SojaCSSLP5*, including 147, 7 and 23 in a total of 177 CSSLs, respectively. 0(144) or 0(147) represent the number of lines with genotype 0 (homozygous *NN1138-2*) are 144 or 147, while 2(3) or 2(0) represent the number of lines with genotype 2 (homozygous N24852) are 3 or 0.

**Figure 3 ijms-22-01559-f003:**
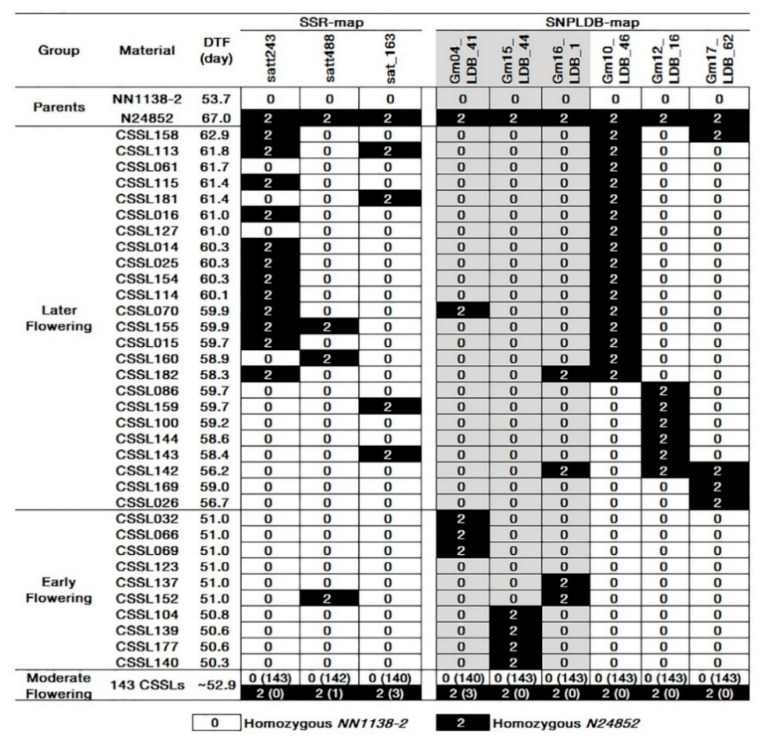
Joint comparisons of CSSLs significantly different from *NN1138-2* in DTF. DTF, days to flowering. SSR-map, the physical map constructed with SSR markers; SNPLDB-map, the physical map constructed with SNPLDB markers. The wild segments on the markers in gray cells are of negative additive effects; the others are of positive additive effects. In 0(143), 0 represents *NN1138-2* genotype while (143) represents 143 CSSLs with genotype 0 in intermediate flowering group. Similarly, 2(3) represents 3 CSSLs with genotype 2 (N24852) in intermediate flowering group.

**Figure 4 ijms-22-01559-f004:**
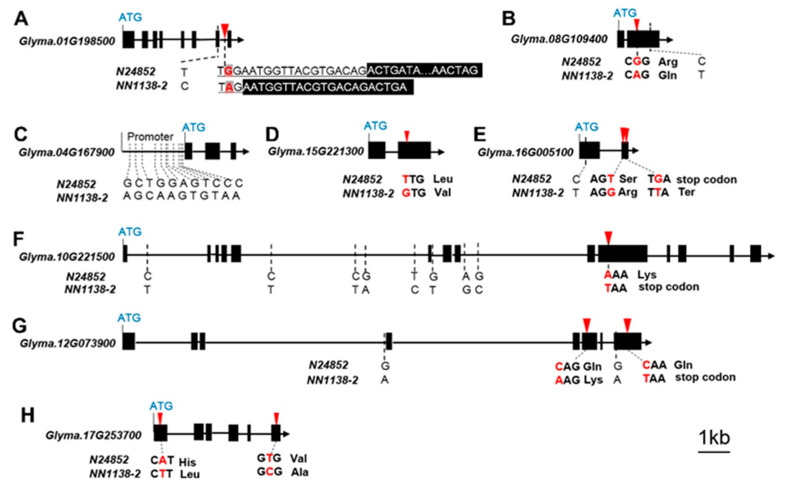
Sequence analysis of SCC and DTF candidate genes between *N24852* and *NN1138-2*. (**A**) and (**B**), are candidate genes of SCC, respectively; (**C**–**H**) are respective DTF candidate genes. The black boxes represent exons; the lines between boxes represent introns; red arrowheads and broken lines represent the variant position. Red nucleotides represent SNP between *N24852* and *NN1138-2,* which are nonsynonymous mutations and marked with red arrowheads.

**Table 1 ijms-22-01559-t001:** SSR and SNPLDB linkage maps of *SojaCSSLP5.*

Chr.	No. Markers		No. Segments		New Segments	No. Broken Sites	Average Length (Mb)		Coverage (%)	
	SSR	SNPLDB	SSR-Map	SNPLDB-Map			SSR-Map	SNPLDB-Map	SSR-Map	SNPLDB-Map
1	6	82	36	72	35	1	19.94	10.08	100	100
2	13	71	41	65	21	3	14.21	8.09	100	100
3	7	50	24	55	29	2	10.05	4.37	100	99.79
4	10	88	55	98	37	6	10.36	4.36	100	100
5	8	49	40	83	39	4	9.59	5.25	100	98.32
6	11	125	68 (1)	199 (1)	120	11	6.77	2.72	100	100
7	6	51	29	54	25	0	12.13	6.22	100	96.30
8	12	93	62	84	22	0	6.38	5.29	100	100
9	9	89	44 (1)	155 (1)	88	23	9.05	3.04	100	100
10	9	60	28	51	22	1	12.72	8.09	100	100
11	10	73	44(1)	104 (3)	45	15	5.38	2.69	100	100
12	8	49	23	43	18	2	11.20	6.67	100	100
13	10	63	35	63(1)	21	7	7.06	4.45	100	100
14	11	48	33	60	27	0	12.49	7.81	100	100
15	6	54	18	47	29	0	12.99	6.30	100	100
16	8	72	46	72	20	6	7.32	4.63	100	100
17	11	68	37	62	25	0	7.48	4.26	100	100
18	8	42	21 (1)	41 (1)	21	0	14.24	8.37	100	100
19	10	78	43	100	54	3	10.11	5.45	100	100
20	9	61	36	59 (2)	23	0	13.37	7.47	100	100
Total	182	1366	763	1568	721	84	10.06	5.16	100	99.74

Chr., chromosome; SSR-map, the physical map constructed with SSR markers; SNPLDB-map, the physical map constructed with SNPLDB markers; New segments, the segment number detected in SNPLDB-map but not found in SSR-map; No. broken sites, number of broken sites in SSR-map due to SNPLDBs. The number in brackets indicated the number of heterozygous segments in column of NO. segments.

**Table 2 ijms-22-01559-t002:** Markers identified for SCC and DTF in SSR-map and SNPLDB-map.

	Segment	Position (bp)	Interval (Mb)	LOD	PVE (%)	ADD	*C_P&G_*	Predicated Gene/QTL
SCC								
SSR	Sat_160	Chr. 1: 51,845,985–54,121,682	2.28				33%	*G*
	AW132402	Chr. 8: 8,795,838–13,551,423	4.76				77%	*I*
SNPLDB	Gm01_LDB_74	Chr. 1: 52,865,890–53,515,092	0.65				100%	*G*
	Gm08_LDB_32	Chr. 8: 8,015,591–8,819,462	0.80				100%	*I*
**DTF**								
SSR	Satt243	Chr. 10: 45,344,616–47,643,217	2.30	22.14	39.76	3.58	100%	*E2*
	Satt488	Chr. 17: 20,123,420–32,039,054	11.92	2.81	3.89	1.56	60%	*New*
	Sat_163	Chr. 18: 2,420,063–3,620,588	1.20	3.99	5.61	1.71	67%	*First flower 21-4*
Total	3				49.25			
SNPLDB	Gm04_LDB_41	Chr. 4: 41,449,035–42,363,602	0.91	2.88	0.77	−0.58	43%	*New*
	Gm10_LDB_46	Chr. 10: 45,288,662–45,566,206	0.28	79.28	67.05	3.68	100%	*E2*
	Gm12_LDB_16	Chr. 12: 5,497,551–5,546,301	0.05	29.46	11.42	2.63	100%	*GmPRR3B*
	Gm15_LDB_44	Chr. 15: 20,056,071–39,957,410	19.9	7.87	2.26	−1.00	100%	*qR1-a-15-2*
	Gm16_LDB_1	Chr. 16: 1–480,323	0.47	7.78	2.23	−0.72	50%	*New*
	Gm17_LDB_62	Chr. 17: 40,650,273–40,782,337	0.13	13.75	4.28	2.07	100%	*New*
Total	6				88.02			

DTF, days to flowering; SCC, seed coat color; SSR, Simple Sequence Repeat; SNPLDB, SNP linkage disequilibrium blocks. Chr., chromosome; LOD, logarithm of odds; PVE, percentage of phenotypic variation explained by individual QTL; ADD, additive effect of the allele from *N24852* (*G. soja*). C_*P&G*_, consistency between phenotype and genotype which is the number of lines with genotype co-segregating with phenotype divided by the number of lines containing the detected wild allele. *G*, the gene related to green seed coat color; *I*, the gene related to black seed coat color.

**Table 3 ijms-22-01559-t003:** Expression patterns of candidate genes related to SCC and DTF.

Locus	Gene	Parents	Tissue (FPKM, Fragments per Kilobase per Million)							
			Leaf	Flower	14 seed	21 seed	28 seed	35 seed	7 pod	21 pod
Gm01_LDB_74	*Glyma.01G198500*	*N24852*	113.96	**8.69**	**14.96**	8.76	7.05	**2.57**	16.18	**11.98**
		*NN1138-2*	109.98	**20.01**	**3.55**	9.02	4.25	**9.49**	24.26	**32.42**
Gm08_LDB_32	*Glyma.08G109400*	*N24852*	**1.52**	**1.68**	**1.80**	**19.09**	**1.60**	**0.11**	0.26	**0.03**
		*NN1138-2*	**6.78**	**5.09**	**9.45**	**39.32**	**6.51**	**4.81**	0.35	**0.73**
Gm04_LDB_41	*Glyma.04G167900*	*N24852*	**330.30**	37.96	**67.54**	16.98	**1.87**	**0.28**	166.90	90.83
		*NN1138-2*	**677.28**	68.83	**22.22**	33.65	**19.00**	**4.47**	109.97	118.47
Gm10_LDB_46	*Glyma.10G221500*	*N24852*	**76.29**	**26.39**	27.96	16.87	**27.73**	6.97	22.28	**143.75**
		*NN1138-2*	**19.85**	**11.59**	19.59	11.02	**4.60**	10.25	16.57	**41.63**
Gm12_LDB_16	*Glyma.12G073900*	*N24852*	**180.77**	16.36	**23.23**	**33.09**	48.69	19.68	**39.38**	**129.56**
		*NN1138-2*	**84.81**	28.13	**47.24**	**15.03**	33.83	26.62	**13.69**	**29.43**
Gm15_LDB_44	*Glyma.15G221300*	*N24852*	**21.13**	148.53	**40.66**	**9.88**	**20.63**	**0.13**	165.97	**6.03**
		*NN1138-2*	**133.21**	121.38	**15.14**	**28.25**	**9.58**	**7.87**	160.57	**33.45**
Gm16_LDB_1	*Glyma.16G005100*	*N24852*	**10.93**	**1.12**	**1.27**	0.00	0.00	0.03	**2.17**	**0.49**
		*NN1138-2*	**0.59**	**0.07**	**0.02**	0.01	0.00	0.00	**0.05**	**0.01**
Gm17_LDB_62	*Glyma.17G253700*	*N24852*	**1.56**	3.79	9.96	**3.73**	2.38	**0.00**	13.10	2.65
		*NN1138-2*	**19.21**	4.87	13.11	**13.04**	3.74	**3.98**	16.70	5.24

In Tissue column, Leaf, collected before flowering; Flower, collected at flowering; 14 seed, 21 seed, 28 seed and 35 seed, the seeds of 14, 21, 28, 35 days after flowering, respectively; 7 pod and 21 pod, the pod of 7 and 21 days after flowering, respectively. The numbers are the FPKM values of gene expression in different tissues. The cells in boldface indicate that the gene expression differs by more than two-fold between *NN1138-2* and *N24852*.

**Table 4 ijms-22-01559-t004:** Annotation of candidate genes for SCC and DTF based on the sequence differences between *NN1138-2* and *N24852.*

Segment	Candidate Gene	Description	Position (bp)	Allelic Variation		Variant Type
				*N24852*	*NN1138-2*	
Gm01_LDB_74	*Glyma.01G198500 (G)*	CAAX amino terminal protease protein	53,229,579	G	A	Stop gain and splice acceptor variant and intron variant
Gm08_LDB_32	*Glyma.08G109400*	Chalcone and stilbene synthase family protein	8,392,915	G	A	Missense variant
Gm04_LDB_41	*Glyma.04G167900*	Light-harvesting chlorophyll-protein complex I subunit A4	11 SNPs in promoter region			
Gm10_LDB_46	*Glyma.10G221500 (E2)*	Gigantea protein (GI)	45,310,798	A	T	Stop gain
Gm12_LDB_16	*Glyma.12G073900*	Gseudo-response regulator 3	5,519,728	C	A	Missense variant
	*(GmPRR3B)*		5,520,945	C	T	Stop gain
Gm15_LDB_44	*Glyma.15G221300*	UDP-glucosyl transferase 73B3	39,953,287	T	G	Missense variant
Gm16_LDB_1	*Glyma.16G005100*	Unknown protein	349,809	T	G	Missense variant
			349,961	G	T	Stop loss
Gm17_LDB_62	*Glyma.17G253700*	UDP-Glycosyltransferase superfamily protein	40,762,395	A	T	Missense variant
			40,766,099	T	C	Missense variant

In description column: CAAX (C = cysteine, A = an aliphatic amino acid, and X = one of several amino acids); UDP, uridine 5’-diphosphate.In allelic variation column: A, T, C and G represent different nucleotide. In variant type column: stop gain and stop loss represent that nucleotide mutation from *N2485*2 to *NN1138-2* generates a premature stop codon in *NN1138-2* or loss stop codon in *N24852*, respectively.

**Table 5 ijms-22-01559-t005:** The effects of wild versus cultivated gene-alleles/haplotypes in *SojaCSSLP5* and germplasm accessions.

Gene	Haplotype	No. Lines	Range		Mean (Day)		*C_P&G/_* Significance
***SojaCSSLP5*** (177 CSSLs)							
*Glyma.01G198500*	**D1**E2	**7**	Green seed coat				100%
	**D2**E2	147	Yellow seed coat				
*Glyma.08G109400*	D2**E1**	23	Black seed coat				100%
	D2**E2**	147	Yellow seed coat				
*Glyma.04G167900*	**F1**H2I2J2K2L2	3	51.0~51.0 (day)		51.0		***
	**F2**H2I2J2K2L2	143	51.2~55.4 (day)		52.8		
*Glyma.10G221500*	F2**H1**I2J2K2L2	13	58.9~61.8 (day)		60.6		***
	F2**H2**I2J2K2L2	143	51.2~55.4 (day)		52.8		
*Glyma.12G073900*	F2H2**I1**J2K2L2	5	58.4~59.7 (day)		59.1		***
	F2H2**I2**J2K2L2	143	51.2~55.4 (day)		52.8		
*Glyma.15G221300*	F2H2I2**J1**K2L2	4	50.3~50.8 (day)		50.6		***
	F2H2I2**J2**K2L2	143	51.2~55.4 (day)		52.8		
*Glyma.16G005100*	F2H2I2J2**K1**L2	2	51.0~51.0 (day)		51.0		**
	F2H2I2J2**K2**L2	143	51.2~55.4 (day)		52.8		
*Glyma.17G253700*	F2H2I2J2K2**L1**	2	56.7~59.0 (day)		57.8		***
	F2H2I2J2K2**L2**	143	51.2~55.4 (day)		52.8		
**Germplasm population** (303 cultivars)							
*Glyma.01G198500*	**D1**	44	Green seed coat			100%	
	**D2**	259	Yellow seed coat				
*Glyma.10G221500*	**H1**I2J2K2L2	11	37.0~68.0 (day)	52.3		***	
	**H2**I2J2K2L2	38	30.3~49.0 (day)	41.5			
*Glyma.12G073900*	H2**I1**J2K2L2	7	45.3~59.7 (day)	49.7		***	
	H2**I2**J2K2L2	38	30.3~49.0 (day)	41.5			
*Glyma.15G221300*	H2I2**J1**K2L2	6	33.0~41.3 (day)	36.9		*	
	H2I2**J2**K2L2	38	30.3~49.0 (day)	41.5			
*Glyma.16G005100*	H2I2J2**K1**L2	47	30.3~63.3 (day)	37.9		**	
	H2I2J2**K2**L2	38	30.3~49.0 (day)	41.5			
*Glyma.17G253700*	H2I2J2K2**L1**	14	38.7~65.5 (day)	52.1		*	
	H2I2J2K2**L2**	38	30.3~49.0 (day)	41.5			

D1, E1, F1, H1, I1, J1, K1 and L1 represent haplotypes of *Glyma.01G198500*, *Glyma.08G109400*, *Glyma.04G167900*, *Glyma.10G221500*, *Glyma.12G073900*, *Glyma.15G221300*, *Glyma.16G005100* and *Glyma.17G253700* consistent to wild soybean *N24852*, respectively. D2, E2, F2, H2, I2, J2, K2 and L2 represent haplotypes of *Glyma.01G198500*, *Glyma.08G109400*, *Glyma.04G167900*, *Glyma.10G221500*, *Glyma.12G073900*, *Glyma.15G221300*, *Glyma.16G005100* and *Glyma.17G253700* consistent to cultivated soybean *NN1138-2*, respectively. **D1**E2 represents the wild haplotype **D1** in E2 genetic background; **F1**H2I2J2K2L2 represents the wild haplotype **F1** in H2I2J2K2L2 genetic background. The same is for the others. *, ** and *** indicate significant at *p* < 0.05, *p* < 0.01 and *p* < 0.001, respectively, in t-tests.

## Data Availability

All data are available upon reasonable request.

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
