# Peer review of "Identifying Wild Versus Cultivated Gene-Alleles Conferring Seed Coat Color and Days to Flowering in Soybean"

_ijms, 2021, doi:10.3390/ijms22041559_

Round 1

Reviewer 1 Report

The paper "Identifying wild versus cultivated gene-alleles conferring seed coat color and days to flowering in soybean" by Cheng Liu et al. is a well designed, well done work with adequate statistical design and analyses, and standard molecular and bioinformatic methods. As such, is a nice piece of work that could be published, given that the Conclusions section is improved in explaining in plain language something about the characteristics of the wild populations and the cultivated plants and the characters that were transferred between them.

For instance, in the Introduction there are some valuable lines describing morphological traits of seeds, stems, and pods, helping the reader to make a picture of the real plants. Something like this would be desirable also in the conclusions. Instead of, the Conclusions are full with concepts which are described more clearly in the Discussion. Also, some other general reference about the CSSL strategy would be useful.

Ad referendum of the Editor, I would suggest this work to be accepted for its publication, after the Conclusions are improved as suggested. 

Author Response

Point 1: The paper "Identifying wild versus cultivated gene-alleles conferring seed coat color and days to flowering in soybean" by Cheng Liu et al. is a well designed, well done work with adequate statistical design and analyses, and standard molecular and bioinformatic methods. As such, is a nice piece of work that could be published, given that the Conclusions section is improved in explaining in plain language something about the characteristics of the wild populations and the cultivated plants and the characters that were transferred between them. For instance, in the Introduction there are some valuable lines describing morphological traits of seeds, stems, and pods, helping the reader to make a picture of the real plants. Something like this would be desirable also in the conclusions. Instead of, the Conclusions are full with concepts which are described more clearly in the Discussion. Also, some other general reference about the CSSL strategy would be useful. Ad referendum of the Editor, I would suggest this work to be accepted for its publication, after the Conclusions are improved as suggested.

Response 1: Many thanks for the comment. The conclusion has been improved, and a revised version has been replaced in line 604-620. This revised version is “Wild soybean (G. soja), characterized with small and black seed, twining stem, pod shattering, etc. is acknowledged to be the wild progenitor of the cultivated soybean (G. max). To reveal the genetic changes from G. soja to G. max, an improved wild soybean CSSL population SojaCSSLP5 with segment heterozygosity reduced and both SSR and SNPLDB markers genotyped was established based on the previous versions obtained through advanced backcrossing. Compared to the low density of SSR-map, the highly densified SNPLDB-map could identify many more wild segments with shorter length in SojaCSSLP5 (1366 vs. 182 markers and 5.06 vs. 10.06 Mb/marker). By using SojaCSSLP5 SNPLDB-map, 2 markers co-segregating with SCC and 6 markers associated to DTF with 88.02% PVE were mapped, among which 3 DTF QTLs were newly detected in the wild soybean. Integrated with parental RNA-seq and DNA resequencing, 2 and 6 candidate genes were predicted based on allele-phenotype coincidence in SojaCSSLP5 for SCC and DTF, respectively. Among these, one SCC candidate gene and 4 DTF candidate genes were newly reported here that might be used to broaden the genetic base of cultivated soybeans. The present results indicated that the mapping resolution of QTLs and qualitative genes was highly improved by using high density SNPLDB-map, and this kind of CSSL-map, if integrated with multiple allele information in germplasm population, is an potential platform in identifying candidate wild vs. cultivated gene-alleles and exploring evolutionary mechanism from wild to cultivated soybeans.”.

Reviewer 2 Report

Liu et al, attempted to identify genes for seed coat and DTF. There are some major points need to be addressed before publication.

To my eyes paper is prepared in a semi-confusing way. T

Introduction: page2, line 48: the concept of Gene, QTL and Allele have been used in a very confusing and sometimes incorrect approaches. These 3 items are different and need to be addressed clearly.

Introduction: page 2, lines 76-80. This section contains incomplete, incorrect and confusing text. For example, E serries are early maturity loci (QTLs). Also, there are E1-E11 which should be addressed clearly.

Results, page 2, line 92: Does a set of 182 RRSs statistically enough to cover the genome with approach?

Results: It is not clear for me how SSR and SNP data have been collective used and analyzed. A clear and detailed explanation is recommended.

Table 2: what “G” and “I” stand for?

Results, page 6, line 174. Authors indicated “When using the SSR-map, we were failed to detect the marker that was co….” cloud you please elaborate more on this part and make it clear in the text. 

 Section 2.3.1, It is not clear for me how Glyam.01G198500, in particular ‘ being selected? Just based on GO? And what about other genes in table 3?

Same comment for table 4 and section 2.3.2.

Page 8, line 245: what is G gene?

General: authors indicated of cloning approaches. For some reason I was not able to see the results and any details in methods section!

Section 2.3.2: in selection process for candidate genes for DTF, I was wondering if long-days vs short-day growth approached have been considered? If not, why? If yes, please elaborate in text.

Methods: For SNP calling, how did authors filter rare alleles?

Methods: please add details about the RNS-seq approach.

Methods: cloning?

Author Response

Response to Reviewer 2 Comments

Comment: Liu et al, attempted to identify genes for seed coat and DTF. There are some major points need to be addressed before publication.

Response: Many thanks. Please see the following answers to the comments.

Point 1: Introduction: page2, line 48: the concept of Gene, QTL and Allele have been used in a very confusing and sometimes incorrect approaches. These 3 items are different and need to be addressed clearly.

Response 1: According to the comment, to make the concept clear in the present study, three pairs of terms were used in gene and QTL mapping studies, i.e. segment (marker) and its haplotypes, qualitative gene and its alleles, QTL and its alleles; while candidate gene and its alleles were used in candidate gene analysis. Here “allele” was used in different cases because it was used in these ways already in literature. However, it can be distinguished from its context. In those cases of necessity, QTL-alleles, gene-alleles, candidate gene-alleles might be used.

This statement has been incorporated in the text in line 62-67 and, according to the above descriptions, the three pairs of terms were corrected in lines of 172, 174, 191, 213, 228, 230, 232, 245, 261, 290, 303, 343, 366, 408, 410, 413, 416, 417, 422, 426, 543, 546, 579, 590, and 593.

Point 2: Introduction: page 2, lines 76-80. This section contains incomplete, incorrect and confusing text. For example, E serries are early maturity loci (QTLs). Also, there are E1-E11 which should be addressed clearly.

Response 2: Thanks. As the days to flowering and days to maturity are two linked traits in soybean, they were put together when we review the genetic bases of the former. Meanwhile, some new references and sentence were added in line 84-95 to fully and clearly address the genetic advancement of days to flowering in soybean. The corresponding text was changed as the following:

“Up to now, a total of 13 genes including E1 E10 [18-26], J [27], Tof12 and Tof11 [28] and 104 QTLs were identified for DTF or its linked trait DTM (days to maturity) (http://www.soybase.org/). Among them, 7 genes, E1 [29], E2 [30], E3 [31], E4 [32], J [33] and Tof12 and Tof11 [28] have been cloned. E1 has impact on DTF in soybean, and appears to be a legume-specific transcription factor that has a putative nuclear localization signal and a B3 distantly related domain [29]. E2 is an ortholog of the Arabidopsis GIGANTEA (GI) gene to delay flowering and maturity [30]. E3 and E4 are two phytochrome A homologs, GmPHYA3 and GmPHYA2, respectively [31,32]. J as the ortholog of Arabidopsis EARLY FLOWERING 3 (ELF3) depends genetically on E1 [34]. Tof12 and Tof11 are two pseudo-response regulator genes and act via LHY homologs to promote expression of the E1 gene and delay flowering under long photoperiods [28]. E9 is GmFT2a, an ortholog of Arabidopsis FLOWERING LOCUS T. GmFT4 is identified as the potential candidate gene for E10 locus [26].”

Point 3: Results: Does a set of 182 SSRs statistically enough to cover the genome with approach?

Response 3: Because of the large genome of soybean, it is hard to construct the CSSL population with high density PCR-based molecular marker, such as SSR. Based on 182 SSRs, SojaCSSLP5 was composed of 177 CSSLs with the wild genome separated into 177 diagonal segments, which could cover the whole genome of wild soybean. But it is not statistically enough for identifying wild gene/QTL-alleles due to too many genes/QTLs might be in a large piece of segment. That is why we used densified SNPLDB markers based on whole genome re-sequencing to further separate the segments for a relatively concise mapping of genes/QTLs. After that, the substituted wild segments in CSSL population could cover also entire wild soybean genome (99.74%) except four genomic regions of Gm03_LDB_25, Gm05_LDB_3, Gm07_LDB_14 and Gm07_LDB_18, but with some wild/cultivated chimera SSR-segments identified and removed.

Point 4: Results: It is not clear for me how SSR and SNP data have been collective used and analyzed. A clear and detailed explanation is recommended.

Response 4: The SojaCSSLP5 was established using 182 SSR markers based on its previous version SojaCSSLP1 which were developed also using SSR markers. In SojaCSSLP5, the heterozygosity of segments was significantly reduced, accounting for only 0.02%. Then the SojaCSSLP5 was re-sequenced with average depth of 3.0×, and 2,567,426 SNPs were identified, from which 1366 SNPLDB marker segment types were obtained. Therefore, two marker systems or two marker-maps were established on a same set of CSSLs. In this case, some SSR segments were separated into several SNPLDB segments, including some wild/cultivated chimera segments.

We have added the explanation in Materials and Methods section in line 493-498 on page 14-15. In addition, the sentence “To accurately assess the genome composition for improving the QTL mapping resolution, whole genome re-sequencing was carried out for SojaCSSLP5” has been inserted in line 523-524 and the sentence of “In this case, the SSR-map and SNP-map were established for a same CSSL population SojaCSSLP5” has been added to line 541-542.

Point 5: Table 2: what “G” and “I” stand for?

Response 5: G” is the gene related to green seed coat color; “I” is the gene related to black seed coat color. The sentence has been inserted in the notes of Table 2 (line 181-182).

Point 6: Results: Authors indicated “When using the SSR-map, we were failed to detect the marker that was co….” cloud you please elaborate more on this part and make it clear in the text.

Response 6: To make it clear, new sentences were inserted in line 192-197 as the following:

“As shown in Figure 2, a total of 10 CSSLs contained the wild segment of Sat_160 in SojaCSSLP5, among which 3 and 3 CSSLs were with black and yellow seed coat, respectively. For AW132402, a total of 14 CSSLs contained the wild segment on the marker in SojaCSSLP5, among which 3 CSSLs were with yellow seed coat and the others with black seed coat. Meanwhile, 12 black seed coat CSSLs carried cultivated segment of AW132402. So the marker types and phenotypes may be associated but not necessarily co-segregated in SSR-map.”.

Point 7: Section 2.3.1, It is not clear for me how Glyam.01G198500, in particular being selected? Just based on GO? And what about other genes in table 3?

Response 7: The basic idea in prediction and primary verification of candidate genes of SCC and DTF were (1) to find all the genes in the identified segment according to SoyBase (http://www.soyase.org), (2) to choose the possible candidate genes according to the two parent’s RNA-seq data in a set of different tissues, (3) to find the most possible candidate genes according to the sequence differences by using high depth genome resequencing of the two parents, at last (4) to annotate the predicted candidate gene based on Gene Ontology (GO). For example, in searching for the candidate gene of G locus, (1) the linked segment of Gm01_LDB_74 was between 52,865,890 bp to 53,515,092 bp on Chr. 01, in which 81 genes were contained according to SoyBase (http://www.soybase.org); (2) among them, Glyam.01G198500 performed significantly different expressions of more than two-folds between two parents among four tissues, including flower, 14seed, 35seed and 21Pod; (3) the genome sequence of Glyma.01G198500 was different between the two parents with the nucleotide G in N24852 mutated to A in NN1138-2 in intron region, which led to an alternative splicing site and generated a premature stop codon; (4) Glyam.01G198500 encodes the CAAX amino terminal protease protein based on GO. Other 7 candidate genes were also provided with the same method as Glyam.01G198500.

To make it clear, the paragraph in line 254-259 has been changed to:

“The basic idea in prediction and primary verification of candidate genes of SCC and DTF were (1) to find all the genes in the identified segment according to SoyBase (http://www.soyase.org), (2) to choose the possible candidate genes according to the two parent’s RNA-seq data in a set of different tissues, (3) to find the most possible candidate genes according to the sequence differences by using high depth genome resequencing of the two parents, and (4) to annotate the predicted candidate gene based on Gene Ontology (GO) analysis.”.

Point 8: Same comment for table 4 and section 2.3.2.

Response 8: The idea of selected candidate genes have been described in Answer to Comment 7. Table 3 shows the difference of candidate genes in expression level, and Table 4 shows the sequence differences of candidate genes between the two parents and its predicted functions.

Point 9: Page 8, line 245: what is G gene?

Response 9: G is the classic qualitative locus conferring green seed coat in soybean. The functional gene is Glyma.01G198500. The word “(Glyma.01G198500)” has been inserted in Line 269.

Point 10: General: authors indicated of cloning approaches. For some reason I was not able to see the results and any details in methods section!

Response 10: The sequences of predicted genes were compared based on the whole genome re-sequencing data of the parents, NN1138-2 and N24852. As Glyma.12G073900 may be the most likely candidate gene of DTF, we also design primers to clone it from NN1138-2 and N24852. Then we compare their sequence differences, which indicated that the C and C nucleotides in the wild soybean N24852 was mutated to A and T in the cultivated soybean NN1138-2, respectively.

In addition, the cloning of Glyma.12G073900 was added in Results section line 338-339 and the cloning method was added in Materials and Methods section line 571-574 as the following:

“To clone and compare the CDS of Glyma.12G073900, the gene-specific primers were designed based on the cDNA sequence of Williams82 v2.1, and used for PCR amplification of each homologous gene from NN1138-2 and N24852. The primers were: 5’TCAAGTGCTTGGGATGTGGA3’and 5’GCCGTACTTCATCTGTCCCG3’.”.

Point 11: Section 2.3.2: in selection process for candidate genes for DTF, I was wondering if long-days vs short-day growth approached have been considered? If not, why? If yes, please elaborate in text.

Response 11: What we used in selection process for candidate genes of DTF was on mainly RNA expression analysis and coincidence analysis between alleles and phenotypic performance. We thought that the different DTF performances under three environments could provide a significant detection between alleles, therefore did not consider to test the materials under long-days vs short-day conditions. To indicate the growth conditions, the expression of “and were all belonged to short-day growth conditions” was inserted in Line 509.

Point 12: Methods: For SNP calling, how did authors filter rare alleles?

Response 12: We did not filter rare SNP in the present paper due to the extreme skewness of allele frequency toward the recurrent parent in SojaCSSLP5. The allele frequency of donor genotype may be 1/177, if there is only one substitution segment existed in SojaCSSLP5.

Point 13: Methods: please add details about the RNA-seq approach.

Response 13: We have added details about the RNA-seq approach in line 559-569 in the Methods and Methods section.

Point 14: Methods: cloning?

Response 14: As in the Answer to Comment 10, the cloning method has been added in Materials and Methods section line 571-574.

Round 2

Reviewer 2 Report

I believe authors did a good job responding to the comments and adjusting the text. In my humble opinion, this manuscript is suitable for publication.